# Magnetic Resonance Imaging Biomarkers of Punding in Parkinson’s Disease

**DOI:** 10.3390/brainsci13101423

**Published:** 2023-10-07

**Authors:** Chenglu Mao, Yang Zhang, Jialiu Jiang, Ruomeng Qin, Qing Ye, Xiaolei Zhu, Jiayong Wu

**Affiliations:** 1Department of Neurology, Nanjing Drum Tower Hospital, Affiliated Hospital of Medical School, Nanjing University, Nanjing 210008, China; maochenglu922@163.com (C.M.); zxy419@163.com (Y.Z.); jiangjialiu@126.com (J.J.); ruomengqin@163.com (R.Q.); yeqingyouxiang@126.com (Q.Y.); zhuquelee@126.com (X.Z.); 2Department of Neurology, Nanjing Drum Tower Hospital, State Key Laboratory of Pharmaceutical Biotechnology and Institute of Translational Medicine for Brain Critical Diseases, Nanjing University, Nanjing 210008, China

**Keywords:** Parkinson’s disease, punding, cortical thickness, white matter tract, network topologies

## Abstract

Punding is a rare condition triggered by dopaminergic therapy in Parkinson’s disease (PD), characterized by a complex, excessive, repetitive, and purposeless abnormal movement, and its pathogenesis remains unclear. We aimed to assess the brain structure alterations related to punding by using multipametric magnetic resonance imaging (MRI). Thirty-eight PD patients (19 with punding and 19 without punding) from the Parkinson’s Progression Marker Initiative (PPMI) were included in this study. Cortical thickness was assessed with FreeSurfer, and the integrity of white matter fiber tracts and network topologies were analyzed by using FMRIB Software Library (FSL) and Pipeline for Analyzing braiN Diffusion imAges (PANDA). PD patients with punding showed a higher apathy score and more severe cortical atrophy in the left superior parietal, right inferior parietal, and right superior frontal gyrus, and worse integrity of the right cingulum cingulate tract compared to those without punding. On the other hand, no significant difference in structural network topologies was detected between the two groups. These data suggest that the specific area of destruction may be an MRI biomarker of punding risk, and these findings may have important implications for understanding the neural mechanisms of punding in PD.

## 1. Introduction

Parkinson’s disease (PD) is a neurodegenerative disease prevalent among the elderly population, with an estimated prevalence of 1.7% in individuals over 65 years of age in China [1]. It is characterized mainly by behavioral symptoms, including bradykinesia, rest tremor, rigidity, and changes in posture and gait, and by nonbehavioral symptoms, such as sleep disorder, olfactory disorder, autonomic nerve dysfunction, and cognitive and mental disorder [2]. Punding, a complex, excessive, repetitive, and purposeless abnormal movement, similar to obsessive–compulsive disorder, involves less complicated actions, such as aimless driving, walking, or cleaning repeatedly, and more complicated actions, such as hobbyism [3]. The incidence of punding varies between 1.4% and 14% worldwide, and this considerable variation in incidence is potentially related to the different treatment practices followed in different countries [3,4,5,6]. Long-term treatment with high doses of levodopa or dopamine receptor agonists (DAs) [6], severe attention deficit hyperactivity disorder, younger age of onset, longer disease duration, and male gender have been suggested as risk factors for punding [7]. However, the neural mechanisms underlying the development of punding in patients with PD remain relatively unclear.

Previous studies have suggested that dopaminergic medications can excessively stimulate the D1 receptor and suppress γ-aminobutyric acid (GABA) neurons, leading to the sensitization of striatal dopaminergic receptors [8,9,10]. Alterations in the plasticity of the dorsal and ventral striatum can increase the tendency of patients toward habitual learning. Additionally, damage to the orbitofrontal cortex may contribute to impaired impulse control and inhibitory function, which are related to punding behaviors [3]. The dopamine-mediated reward mechanism is closely related to punding, as thinning of the dorsolateral prefrontal cortex can inhibit the reward system by damaging the projection fibers [11]. Alterations in the white matter microstructure of various tracts, including the right pedunculopontine tract, uncinate fasciculus, genu of the corpus callosum, and left parahippocampal tract, can also impede the transmission of the reward system [12]. To the best of our knowledge, few studies have investigated the relationship between cortical thickness and integrity of the white matter tract with regard to punding in patients with PD.

In recent years, there has been mounting evidence supporting the concept that structural connectivity serves as the material foundation for network connectivity within the brain [13]. The brain uses a topological network comprising nodes (reflecting cortical atrophy) and edges (reflecting white matter fibers) to regulate its structural organization and integrity. It is crucial to demonstrate this topological network to comprehend how different brain regions collaborate to process information and generate behavior and to determine how alterations in this network may be related to various neurological and psychiatric disorders. A technique to study the structural connectivity of the brain is diffusion tensor imaging (DTI), which uses magnetic resonance imaging (MRI) to visualize the diffusion of water molecules in the white matter of the brain [14]. The chaotic topological network features of the brain have been proposed as a potential mechanism for neurological disorders such as Alzheimer’s disease, stroke, and multiple sclerosis [15,16,17]. However, the extent to which punding affects the degree of isolation in a structured network topology and the relationship between punding and the topological network have not yet been fully explored. Therefore, in the present study, we aimed to reveal the correlation of cortical atrophy and white matter fiber damage with punding and to determine the pattern of relationship between the white matter topological network and punding in patients with PD.

## 2. Materials and Methods

### 2.1. Participants

The data of all included subjects were collected from Parkinson’s Progression Markers Initiative (PPMI, http://www.ppmi-info.org/ accessed on 8 May 2021), an international, longitudinal, multicenter study focused on biomarker assessment of recently diagnosed Parkinson’s disease (PD) patients. All subjects included in this study underwent dopamine transporter (DAT) PET-CT prior to their enrollment in PPMI and were definitively diagnosed with PD. Exclusion criteria for PPMI included atypical PD syndrome, other neurological disorders, and the use of drugs considered to interfere with DaTscan imaging within the past 6 months. Punding was assessed using the Questionnaire for Impulsive–Compulsive Disorders (QUIP), which evaluates seven items: gambling, hypersexuality, buying, eating, walking or driving, punding, and hobbyism. A score of 1/2 point was assigned if the judgment of punding or hobbies was “yes”, and subjects were included in the PD-with-punding group (PD-punding). Conversely, a score of 0 was assigned to subjects who did not exhibit punding or hobbyism and were included in the PD-no-punding group (PD-no-punding). After excluding subjects with incomplete data and those who lost data during image postprocessing, 19 PD patients with punding were finally included in this study. A total of 19 PD patients without punding matched by age, gender, education, PD duration, Hoehn and Yahr scale (H&Y), Unified Parkinson’s Disease Rating Scale part III (UPDRS-III), rigidity score, tremor score, and Montreal Cognitive Assessment (MoCA) were selected. Additionally, all subjects were evaluated for depression, anxiety, apathy, and fatigue. Demographic and clinical data were summarized in Table 1. All subjects underwent T1 and DTI MRI sequences.

### 2.2. Clinical and Neuropsychological Assessments

The UPDRS-III, rigidity score, and tremor score were employed to evaluate the gravity of behavioral symptoms in patients with Parkinson’s disease. The diagnosis and severity of nonbehavioral symptoms, including depression, anxiety, apathy, and fatigue, were assessed through UPDRS part I. Additionally, the Hoehn and Yahr scale was utilized to assess the overall severity of the disease in PD patients, while the MoCA was employed to evaluate the mental statements.

### 2.3. MRI Acquisition

All three-dimensional T1-weighted MRI scans were performed on a Siemens 3.0 Tesla MR scanner. The acquisition parameters were as follows: Field Strength = 3.0 tesla; Flip Angle = 9.0 degree; Matrix X = 240.0 pixels; Matrix Y = 256.0 pixels; Matrix Z = 176.0; Mfg Model = TrioTim; Pixel Size X = 1.0 mm; Pixel Size Y = 1.0 mm; Pulse Sequence = GR/IR; Slice Thickness = 1.0 mm; TE = 3.0 ms; TI = 900.0 ms; TR = 2300.0 ms.

All DTI images were acquired with a Siemens 3.0 Tesla MR scanner. The acquisition parameters were as follows: Field Strength = 3.0 tesla; Flip Angle = 90.0 degree; Matrix X = 1044.0 pixels; Matrix Y = 1044.0 pixels; Matrix Z = 65.0; Mfg Model = TrioTim; Pixel Size X = 2.0 mm; Pixel Size Y = 2.0 mm; Pulse Sequence = EP; Slice Thickness = 2.0 mm; TE = 88.0 ms; TR = 700.0 ms.

### 2.4. Image Processing

MRI analysis was performed by an experienced researcher who was blind to the subject identity.

FreeSurfer software version 5.1 (http://surfer.nmr.harvard.edu accessed on 10 June 2021) was used to perform volumetric measurements of the cortex and subcortex (including the hippocampus and basal ganglia) from T1-weighted images. The estimated Total Intracranial Volume (eTIV) was calculated as the sum of the volume of gray matter, white matter, and cerebrospinal fluid. Preprocessing of FreeSurfer: (1) remove the skull and scalp, (2) register images into standard space, (3) segment white matter and gray matter. After preprocessing, we reinspected the results of each subject. Cortical thickness was determined by measuring the distance between the gray and white matter surfaces at each vertex of the reconstructed cortical mantle. Monte Carlo simulation correction was used for multiple comparison corrections, with a vertex-wise cluster-forming threshold of 2 (*p* < 0.01) and a cluster-wise *p* < 0.05.

FMRIB Software Library version 5.0 (FSL, https://fsl.fmrib.ox.ac.uk accessed on 10 June 2021) was used to preprocess diffusion tensor images. Steps included: (1) remove nonbrain structures; (2) posit front and rear joint; (3) affine image registration and eddy current program. Then we used Pipeline for Analyzing braiN Diffusion imAges version 1.3.1 (PANDA, http://www.nitrc.org/projects/panda accessed on 10 June 2021) [18] to get 20 white matter tracts and obtain fractional anisotropy (FA) and mean diffusivity (MD) values in each subject. A *p*-value of 0.05 with false discovery rate (FDR) correction for multiple comparisons was applied to compare the mean FA and MD.

Following DTI preprocessing, PANDA was utilized to construct the network, which included the definition of nodes and edges. Firstly, AAL 90 was employed to segment the brain into 90 regions (excluding the cerebellum) and transformed into MNI standard space using the DTI sequence. Secondly, each node was taken as the seed point, and fiber tracking was performed with a threshold of FA greater than 0.2. Tracking was stopped when the included angle between two fibers was greater than 45 degrees. Thirdly, the fiber number (FN) network, average FA value network, and average fiber length network were generated. An FN threshold greater than 3 was applied to network binarization. Lastly, the Graph-theoretical Network Analysis Toolkit version 2.0 in MATLAB (GRETNA, https://github.com/sandywang/GRETNA accessed on 22 June 2021) was employed to calculate both global and nodal topological network characteristics.

### 2.5. Network Parameters Analysis

Network parameters comprise both global and nodal characteristics. Global characteristics include clustering coefficient (Cp), characteristic path length (Lp), and global efficiency. Nodal efficiency is included in nodal characteristics. The clustering coefficient affects the propagation dynamics of the network, whereby a larger average clustering coefficient, with other parameters held constant, results in slower propagation. Characteristic path length refers to the shortest path between two nodes, whereby the shorter the path, the better the brain can integrate information. Global efficiency reflects the overall integration capability of the network and is the average of the reciprocal shortest path between all node pairs. Node efficiency is the reciprocal of the characteristic path length.

Regular networks have a higher clustering coefficient and longer characteristic path length, while random networks have a lower clustering coefficient and shorter characteristic path length. The network with both a high clustering coefficient and short characteristic path length is known as a small-world network (σ), which represents an economical and efficient transmission of information. σ > 1 denotes a small-world network attribute, and the larger the value, the stronger the attribute.

### 2.6. Data Analysis

Descriptive analyses were conducted to depict the demographic, clinical, and cognitive characteristics of the participants. The normality of all continuous variables was assessed before conducting comparisons. Categorical variables were compared using the chi-square test. The independent samples *t*-test and Mann–Whitney test were performed to compare continuous variables between PD-punding and PD-no-punding groups. The Statistical Product and Service Solutions (SPSS, version 21) software package was employed for these analyses. The significance level was set at *p* < 0.05.

## 3. Results

### 3.1. Demographical Data

Based on the matching procedure, the two groups exhibited similarity in terms of age, gender, education, PD duration, H&Y, UPDRS-III, rigidity score, tremor score, and MoCA. The mean punding duration was 46.8 ± 26.6 years. There were no significant differences in depression, anxiety, fatigue scores, and total intracranial volume between the two groups, but the PD-punding group exhibited higher scores in pathological measures of apathy compared to the PD-no-punding group (*p* < 0.05, Table 1).

### 3.2. Cortical Thickness

Compared with PD-no-punding patients, PD-punding patients showed significantly reduced cortical thickness in the left superior parietal gyrus, the right inferior parietal gyrus, and the right superior frontal gyrus after Monte Carlo simulation correction, with age, sex, years of education, course of disease, HY grade, and apathy as the covariable (Figure 1 and Table 2). However, there was no significant difference in the volume of basal ganglia, hippocampus, and eTIV between these two groups.

### 3.3. White Matter Tracts

Taking the apathy score as a covariate, after Bonferroni correction, FA in the right cingulum cingulate (CCing_R) tract was significantly decreased in PD-punding patients compared to PD-no-punding patients (Figure 2 and Table 3). However, there were no significant differences in other tracts between the two groups.

### 3.4. Network Parameters

Global and nodal characteristics did not demonstrate significant differences between the groups, but the PD-punding group had a tendency to decrease in Cp, global efficiency, and local efficiency, as well as to increase in Lp. Both groups displayed small-world networks (σ > 1), but there was no significant difference (*p* > 0.05, Table 4).

## 4. Discussion

To the best of our knowledge, the present study is the first to comprehensively analyze the cortex, white matter fibers, and network topologies of PD patients with punding by using multipametric MRI. Our results indicate that PD patients with punding have thinner cortex in the right inferior parietal gyrus, right superior frontal gyrus, and left superior parietal gyrus and decreased integrity of the right cingulum cingulate as compared to PD patients without punding. These structural changes may reflect the disconnect and imbalance between reward networks in punding patients and may play a role in the development of punding behavior in PD patients. Improved understanding of the underlying mechanisms of punding might help enhance therapeutic strategies for these important disorders.

Apathy, characterized by a syndrome of loss or reduction of motivation compared to an individual’s previous state, occurs in 39.8% of patients with Parkinson’s disease [19]. According to previous research, PD patients with punding have a higher risk of developing psychiatric disorders such as depression and apathy [20,21,22,23]. Consistent with these findings, our present study revealed that PD patients with punding exhibit more apathy than the control group. Apathy is thought to be closely associated with the reward circuitry [24,25]. Previous studies also reported that PD patients with apathy exhibit gray matter loss in the parietal cortex, lateral prefrontal cortex, and orbitofrontal cortex, as well as white matter dysfunction in the cingulum cingulate, uncinate fasciculus, superior longitudinal fasciculus, and cingulum [12,26]. The comorbidity of apathy and impulsivity has already been reported in studies. We can understand that although punding and apathy are opposite in behavior, punding is highly sensitive to rewards, and apathy is not as sensitive to rewards. Patients with apathy face a greater brain energy cost to evaluate costs and benefits when making decisions [27]. That is, when there is no direct external reward, these patients may be less motivated, leading to behavioral apathy. However, these patients may be overly motivated, which can lead to punding [28].

The reward system comprises several regions, including the anterior cingulate cortex, orbitofrontal prefrontal cortex, ventral striatum, ventral globus pallidus, dorsal prefrontal cortex, amygdala, hippocampus, and thalamus. The corticobasal ganglia circuit is the core of the reward system [29]. The frontostriatal circuit has been extensively studied for its role in impulse control disorders (ICDs) [30], and it is known to regulate dopamine-induced repetitive behaviors [31]. In our present study, we observed cortical thinning of the right superior frontal cortex in PD patients with punding as compared to that in controls; this cortical thinning could lead to the abnormal functioning of the reward system and the subsequent abnormal behavior. The parietal lobe is thought to be involved in higher-level sensory input processes, multisensory and sensorimotor integration, intention, and visual integration of the external space and body [32]. In patients with punding, damage to the parietal lobe results in enhanced intention and behavior control disorders. Basal ganglia are affected in the early stages of PD, as overstimulation of the dopamine D1 receptor increases amygdala volume following treatment with dopaminergic medications [33]. Additionally, functional connectivity differences in the amygdala and habenula have been noted between PD patients with punding and those without any impulsive–compulsive behavior (ICB); however, no volumetric differences were observed [34]. Similarly, in our study, we found no significant differences in basal ganglia volume between the PD-without-punding and PD-with-punding groups. We speculate that alterations in the functional connectivity at the micro scale of the amygdala may not be sufficient to affect its structural performance at the macro scale.

On the basis of MRI findings, abnormalities in the cingulum tract have been reported in various neurological and psychiatric disorders, including schizophrenia, depression, mild cognitive impairment, and Alzheimer’s disease [35]. The cingulate tract is an important white matter pathway that integrates reward and connects brain regions involved in cognitive and executive control, such as the frontal nucleus, parietal nucleus, medial temporal nucleus, subcortical nucleus, and cingulate gyrus [36]. The anterior cingulate cortex, which monitors motor errors and conflicts, encodes the total value of immediate gains [37]. In our present study, we observed significant damage to the cingulate gyrus in PD patients with punding, which may have affected their executive control ability. Interestingly, previous studies using [18F]fluorodopa-PET have indicated increased dopaminergic function in PD patients with ICDs, thus suggesting that these patients have stronger responses to rewards and decreased inhibitory control [38]. Moreover, higher fractional anisotropy (FA) in the anterior corpus callosum, the posterior limb of the right internal capsule, the right posterior cingulate gyrus, and the right thalamus has been observed in PD patients with ICDs, thus indicating more intact reward circuits than in patients without ICDs [39].

We also investigated the relationship between the cortex and white matter tracts by studying brain network properties. In recent years, the brain has been considered a comprehensive map of edges and nodes. Several brain regions, including the superior parietal cortex and the superior frontal cortex, are believed to have high “betweenness centrality” [13]. Network parameters such as characteristic path length, clustering coefficient, and global and local efficiency reflect the topological and functional network characteristics of the brain [40]. In our study, both PD-with-punding and PD-without-punding patients showed small-world characteristics of the brain network, thus suggesting efficient information transmission in both groups. However, compared to PD patients without punding, PD patients with punding showed a tendency for a decrease in their comprehensive processing ability. As the disease progresses, the differences in brain network characteristics between punding and non-punding patients may become more prominent.

The occurrence of punding is not the result of a single factor [41], and previous studies have mostly based on a single imaging structure. We have explored the mechanism from the perspective of connecting the whole brain structure in multiple dimensions, providing a prerequisite for clarifying the pathological mechanism of punding and predicting the occurrence of punding. Clinically, reducing the dose of dopaminergic drugs can improve this abnormal behavior, but the primary symptoms of Parkinson’s disease cannot be controlled [42]; amantadine has also been reported to improve stereotypical behaviors that do not respond to reduced dopamine replacement therapy but may induce or exacerbate psychosis [43]. Repetitive transcranial magnetic stimulation (rTMS) is a recognized noninvasive treatment for neurological disease. At present, there are few studies on the treatment of punding with TMS, and only one study that evaluated that TMS stimulation of the dorsolateral prefrontal lobe can temporarily improve the motor symptoms of punding [44]. Our study provides evidence for new targets for subsequent TMS therapy.

## 5. Limitations

To gain a more complete understanding of the pathological mechanisms of punding, future studies should include analyses of both structural and functional networks. A limitation of our present study was the lack of a healthy control group. Therefore, larger studies with healthy controls are required to further analyze the occurrence and development of punding.

## 6. Conclusions

In summary, our study analyzed MRI biomarkers closely related to punding from a noninvasive imaging perspective and suggested that cortical thinning in the frontal and parietal lobes and impairment of white matter microstructure in the right cingulum cingulate tract are associated with the development of punding in PD patients. Furthermore, we also found a high incidence of apathy in punding patients, which we considered to be related to the reward mechanism and decreased sensitivity. These alterations provide insights into the underlying mechanisms of punding and may serve as potential markers for identifying PD patients at risk of developing punding, potentially providing a new direction for subsequent treatment. However, the pathogenesis of punding in PD patients remains unclear, and its mechanism requires further exploration.

## Figures and Tables

**Figure 1 brainsci-13-01423-f001:**
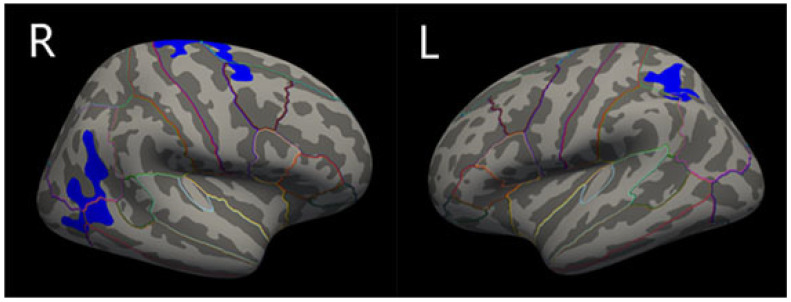
Schematic representations of regional cortical thinning in PD-no-punding and PD-punding groups. Regions of cortical thinning were shown in blue: the right inferior parietal gyrus, the right superior frontal gyrus, and the left superior parietal gyrus. Areas survive the threshold of *p* < 0.05 FDR corrected. Brain regions are divided by lines of different colors.

**Figure 2 brainsci-13-01423-f002:**
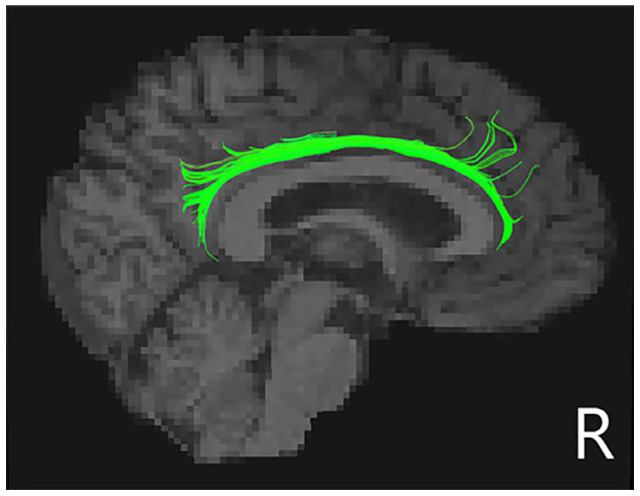
Schematic representations of the decreased right cingulum cingulate tracts in PD-punding groups. Regions of white matter tract was shown in green in the right cingulum cingulate tracts.

**Table 1 brainsci-13-01423-t001:** Characteristics of the study subjects.

	PD-No-Punding (*n* = 19)	PD-Punding (*n* = 19)	Statistics	*p*-Value
Age, year	62.0 ± 8.2	59.1 ± 8.1	1.117	0.271 ^a^
Male, *n*	12 (63.2%)	12 (63.2%)	0.000	1 ^b^
Education, year	14.0 (18.0, 18.0)	13.0 (15.0, 18.0)	129.500	0.129 ^c^
PD duration, month	32.2 ± 21.7	46.8 ± 26.6	−1.864	0.070 ^a^
H&Y	1.0 (2.0, 2.0)	2.0 (2.0, 2.0)	144.500	0.215 ^c^
UPDRS-III	21.1 ± 9.6	23.3 ± 9.5	−0.697	0.491 ^a^
Rigidity	2.0 (3.0, 5.0)	3.0 (4.0, 6.0)	147.000	0.320 ^c^
Tremor	5.2 ± 4.6	4.4 ± 3.8	0.621	0.539 ^a^
MoCA	26.0 (28.0, 29.0)	23.8 (28.0, 30.0)	161.000	0.758 ^c^
Depression	0.3 (0, 1)	0.5 (0, 1)	166.000	0.595 ^c^
Anxious	0.3 (0, 1)	0.7 (0, 1)	138.000	0.159 ^c^
Apathy	0.1 (0, 0)	0.5 (0, 1)	131.500	0.034 ^c^
Fatigue	0.6 (0, 1)	1.2 (0, 2)	139.000	0.052 ^c^
eTIV (mm^3^)	1,619,789.102 ± 139,915.239	1,553,787.163 ± 141,391.857	1.446	0.157 ^a^

Values conforming to the normal distribution are expressed as means ± standard deviations. Values that do not conform to the normal distribution are expressed by the median and interquartile spacing. UPDRS-III: Unified Parkinson’s Disease Rating Scale scores-III; H&Y: Hoehn and Yahr scale; MoCA: Montreal Cognitive Assessment; eTIV: estimated Total Intracranial Volume. ^a^: *p*-value of independent *t*-test. ^b^: *p*-value of chi-squared test. ^c^: *p*-value of Mann-Whitney test.

**Table 2 brainsci-13-01423-t002:** Cortical thickness change in PD with punding and PD without punding.

	PD-No-Punding (*n* = 19)	PD-Punding (*n* = 19)	*p*-Value
L_superior parietal (mm^2^)	2.294 ± 0.082	2.120 ± 0.092	0.000 *
R_inferior parietal (mm^2^)	2.570 ± 0.107	2.357 ± 0.114	0.000 *
R_superior frontal (mm^2^)	2.627 ± 0.142	2.389 ± 0.229	0.000 *

Values are means ± standard deviations. L: left. R: right. *: multiple comparisons of Monte Carlo simulation correction.

**Table 3 brainsci-13-01423-t003:** Diffusion Tensor MRI metrics of white matter tracts in PD with punding and PD without punding.

	Tract	PD-No-Punding (*n* = 19)	PD-Punding (*n* = 19)	*p*-Value
FA	R_CCing	0.500 ± 0.027	0.470 ± 0.026	0.006 *
	L_IFOF	0.458 ± 0.018	0.446 ± 0.018	0.039 ^a^
	R_SLF	0.534 ± 0.041	0.507 ± 0.046	0.023 ^a^
MD (×10^−3^ mm^2^ s^−1^)	R_CCing	0.692 ± 0.032	0.720 ± 0.041	0.038 ^a^
	R_UF	0.793 ± 0.044	0.825 ± 0.044	0.010 ^a^

Values are means ± standard deviations. Only values reporting significant differences between groups were reported. FA: fractional anisotropy; R_CCing: right cingulum cingulate; L_IFOF: left inferior fronto-occipital fasciculus; MD: mean diffusivity; R_SLF: right superior longitudinal fasciculus; R_UF: right uncinated fasciculus. *: multiple comparisons of Bonferroni correction. ^a^: *p*-value of independent *t*-test without multiple comparison correction.

**Table 4 brainsci-13-01423-t004:** Network parameters between two groups.

Network Parameters	PD-No-Punding (*n* = 19)	PD-Punding (*n* = 19)	*p*-Value
σ	3.796 ± 0.343	3.718 ± 0.335	0.484 ^a^
Lp	5.444 ± 0.364	5.525 ± 0.332	0.478 ^a^
Cp	0.280 ± 0.025	0.271 ± 0.028	0.297 ^a^
global efficiency	0.184 ± 0.012	0.182 ± 0.011	0.474 ^a^
local efficiency	0.267 ± 0.019	0.256 ± 0.021	0.085 ^a^

Values are means ± standard deviations. σ: small-world network; Cp: clustering coefficient; Lp: characteristic path length. ^a^: *p*-value of independent *t*-test.

## Data Availability

Source data of this study were from Parkinson’s Progression Markers Initiative (PPMI, http://www.ppmi-info.org/ accessed on 8 May 2021).

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
