# Peer review of "Magnetic Resonance Imaging Biomarkers of Punding in Parkinson’s Disease"

_brainsci, 2023, doi:10.3390/brainsci13101423_

Round 1

Reviewer 1 Report

The article is very interesting and very well written and designed. Only a few doubts remain from the analyses. Were the morphometric analyzes carried out using the FS Qdec tool? I think it would be interesting to include this information, because if it was outside the tool, the metrics would have to be normalized by the intracranial volume and I think this detail was not clear. Another suggestion is to verify the influence of apathy and fatigue, in this result, since they were demographic data and presented and tended to present statistical difference in the groups. And finally, to generalize the data as neuroimaging markers, I think it would be interesting to analyze the correlation between the parameters of morphology, tractography and structural connectivity, associated with clinical practice, it would be interesting to see this data by association or regression. The term multimodal for this study is not the most coherent, because in reality you are using different parameters and sequences of a single technical modality, MRI and not more than one modality, I think the most coherent is the use of multipametric analysis.

Adequate

Reviewer 2 Report

Review of a manuscript “Magnetic Resonance Imaging Biomarkers of punding in Parkinson’s disease” by Chenglu Mao and coauthors submitted to “Brain Sciences”.

Parkinson’s disease is a prevalent and severe neurodegenerative disease second after Alzheimer’s for which there is not medication preventing the course of the disorder.  A small fraction of Parkinson’s disease patients have punding,-  a complex, excessive, repetitive, and purposeless abnormal movement, which is not investigated in detail. Punding is associated with dopaminergic therapy and deserves further study. The authors presented new data on the correlation of cortical atrophy and white matter fiber damage with punding and to determine the pattern of association between the white matter topological network and punding in Parkinson’s disease patients. This is an important area of biomedical research and the manuscript will be interesting for the readers of the journal. The following corrections and additions should be made.

Abstract

Lines 12-13: “Punding is a quite rare condition complicating triggered by dopaminergic therapy in Parkinson’s disease (PD).” This sentence is awkward and should be reshaped. Do the authors want to say “Punding is a quite rare condition triggered by dopaminergic therapy in Parkinson’s disease (PD”?

Introduction

Lines 32-34: After the sentence “It is characterized mainly by behavioral symptoms, including bradykinesia, rest tremor, rigidity, and changes in posture and gait, and by nonbehavioral symptoms such as sleep disorder, olfactory disorder, autonomic nerve dysfunction, and cognitive and mental disorder” the authors should add a reference on a recent review:”Biomarkers in Parkinson’s Disease”. Chapter in a book Peplow PV, et al., eds. Neurodegenerative Diseases Biomarkers. 2022. Neuromethods, v. 173. pp 155-180. Humana, New York, NY.  https://link.springer.com/protocol/10.1007/978-1-0716-1712-0_7

Lines 45-47:” Previous studies have suggested that dopaminergic medications can excessively stimulate the D1 receptor and suppress γ-aminobutyric acid (GABA) neurons, leading to the sensitization of striatal dopaminergic receptors” After this sentese a reference should be added.

Results

Lines 178-179: ”The PD-punding group exhibited higher scores in pathological measures of apathy compared to the PD-no-punding group (p < 0.05, Table 1).” This observation should be discussed in more details in Discussion section. How is it associated with other symptoms and what may be its significance?

Discussion

Lines 229-230 “These findings suggest that these specific gyri and white matter fibers may play a role in the development of punding behavior in PD patients.” The authors should present in more detail their interpretation (at least, hypothetical) how these gyri and white matter fibers play a role in the development of punding behavior

Conclusion

 Lines 294-295. “These alterations provide insights into the mechanisms underlying punding and might serve as potential markers to identify PD patients at risk of developing punding” The authors should add more details about the nature of markers they mean.

Overall, interesting new data presented in the manuscript should be supplemented with more details and interpretations.

Reviewer 3 Report

Dear Authors,

Your article is interesting and has a good presentation, but lacks some aspects in order to be outstanding.

For instance:

It would be beneficial to add a small definition of "punding" in the first sentence of the abstract. Also, the link in the abstract would be better to be removed. The conclusion in the abstract should be enhanced.

The prevalence of PD worldwide should be added in the first two lines. The intentions of this study should be clearly stated in the Introduction section.

In the Material and Methods' section, the inclusion and exclusion criteria should be more clearly stated.

The Results are very well described, but at least one more figure is needed.

The Limitations of your study should be stated in a different chapter. 

Can you add a paragraph in the Discussions' chapter in which to add the significance of your study for the medical- clinical area and also for the patients?

The Conclusions are incomplete for a study of this calibre.

Also, the references bring notoriety to your article, please add more knowledge from the literature.

Round 2

Reviewer 1 Report

All suggestions given to the authors were adopted and the paper is now much clearer and more detailed for the reader to have a greater understanding of the subject. 

Adequate